# Adipose Stromal Cell Spheroids for Cartilage Repair: A Promising Tool for Unveiling the Critical Maturation Point

**DOI:** 10.3390/bioengineering10101182

**Published:** 2023-10-12

**Authors:** Azzurra Sargenti, Simone Pasqua, Marco Leu, Laura Dionisi, Giuseppe Filardo, Brunella Grigolo, Daniele Gazzola, Spartaco Santi, Carola Cavallo

**Affiliations:** 1CellDynamics iSRL, 40136 Bologna, Italy; azzurra.sargenti@celldynamics.it (A.S.); simone.pasqua@celldynamics.it (S.P.); dionisi.laura@libero.it (L.D.); daniele.gazzola@celldynamics.it (D.G.); 2abc biopply ag, 4500 Solothurn, Switzerland; marco.leu@biopply.com; 3Applied and Translational Research (ATR) Center, IRCCS Istituto Ortopedico Rizzoli, 40136 Bologna, Italy; ortho@gfilardo.com; 4Laboratorio RAMSES, IRCCS Istituto Ortopedico Rizzoli, 40136 Bologna, Italy; brunella.grigolo@ior.it; 5Institute of Molecular Genetics “Luigi Luca Cavalli-Sforza”, Unit of Bologna, CNR, 40136 Bologna, Italy; 6IRCCS Istituto Ortopedico Rizzoli, 40136 Bologna, Italy

**Keywords:** adipose stromal cells, spheroids, chondrogenesis, mass density, deep imaging

## Abstract

Articular cartilage lacks intrinsic regenerative capabilities, and the current treatments fail to regenerate damaged tissue and lead only to temporary pain relief. These limitations have prompted the development of tissue engineering approaches, including 3D culture systems. Thanks to their regenerative properties and capacity to recapitulate embryonic processes, spheroids obtained from mesenchymal stromal cells are increasingly studied as building blocks to obtain functional tissues. The aim of this study was to investigate the capacity of adipose stromal cells to assemble in spheroids and differentiate toward chondrogenic lineage from the perspective of cartilage repair. Spheroids were generated by two different methods (3D chips vs. Ultra-Low Attachment plates), differentiated towards chondrogenic lineage, and their properties were investigated using molecular biology analyses, biophysical measurement of mass density, weight, and size of spheroids, and confocal imaging. Overall, spheroids showed the ability to differentiate by expressing specific cartilaginous markers that correlate with their mass density, defining a critical point at which they start to mature. Considering the spheroid generation method, this pilot study suggested that spheroids obtained with chips are a promising tool for the generation of cartilage organoids that could be used for preclinical/clinical approaches, including personalized therapy.

## 1. Introduction

Articular cartilage presents a limited repair capacity due to its poor cellularization [1] and the absence of vascularization [2]. Thus, when cartilage lesions occur because of trauma or degenerative disease, these can evolve into osteoarthritis, causing cartilage and bone degradation, pain, and loss of joint function, impairing significantly patient quality of life [3,4]. The existing clinical approaches, like mosaicplasty, debridement, microfracture, and autologous chondrocyte implantation (ACI), fail to restore damaged tissue, offering only temporary pain relief and no long-term clinical solution [5,6,7]. These limitations have prompted the field of tissue engineering and regenerative medicine (TERM), which seeks to restore, replace, or regenerate tissues and organs using a multidisciplinary approach, including non-cellular therapy [8]. Among the new strategies investigated for developing innovative regenerative therapies, the creation of 3D constructs has emerged using cellular spheroids as building blocks to restore new tissues. 

Compared to 2D, the 3D architecture of spheroids can mimic the natural environment and the hierarchical tissue structures by recapitulating embryonic processes in vitro [9] and improving cell biological and metabolic functions [10]. Spheroids allow for optimizing intracellular signaling, leading to enhanced cell viability, protein secretion, extracellular matrix (ECM) production [11,12], and improving stem cell differentiation processes [13]. Spheroids can be obtained from several cell sources. However, in regenerative medicine, there has been an increased interest in the use of mesenchymal stromal cells (MSCs) due to their regenerative and multipotential properties [14]. In particular, adipose-derived stromal cells (ADSCs) displayed a high chondrogenic potential [15,16,17], and the capacity to recapitulate cartilage formation [18], and the implantation of MSC-spheroids in articular cartilage defects of rabbits also showed to induce the production of hyaline-like cartilage [19].

Despite the initially straightforward self-assembly process, a thoroughly established methodology becomes indispensable to produce a significant quantity of spheroids featuring intricate interactions, distinct architecture, and spherical geometry. Currently, there are several approaches to obtaining spheroids with specific characteristics, such as the hanging drop technique, gel embedding, magnetic levitation, and spinner culture [20]. These techniques can obtain spheroids with similar dimensions, but they are not able to generate equally sized cell aggregates [21,22]. In this light, micro-structured surfaces composed of multiple microcavities with a well-defined diameter and geometry represent a powerful approach to guide and control the formation of spheroids, enabling the creation of 3D structures that closely mimic natural tissue shapes and sizes [23,24]. Moreover, a well-defined spheroid structure and dimension could be involved in the chondrogenic maturation process. This typically requires several weeks to months for the production and deposition of extracellular matrix, which is essential for the formation of functional cartilage tissue, providing mechanical support, and maintaining the chondrogenic phenotype.

However, despite the improvements in 3D structure generation, new methods to characterize spheroids in terms of morphological and physical parameters are needed. The dynamic growth of spheroids is influenced by several factors, with mass density being a significant determinant. Mass density profoundly affects the cellular arrangement within the spheroid, the establishment of nutrient gradients, and the efficacy of cell–cell and cell-extracellular matrix interactions [25]. Mass density and its correlation with size have been previously established as pivotal factors influencing various biological phenomena, such as cell permeation [26] and drug activity [27]. In this light, the comprehension of the complex relationship between mass density and dimension could be essential for understanding their role in the growth and differentiation of spheroids. 

Thus, the aim of this pilot study was to evaluate the capacity of human ADSCs to assemble in spheroids and to differentiate into chondrogenic lineage from the perspective of their use for cartilage repair. Spheroids were obtained using means of two different cultured methods: the 3D CoSeedis™ Chip 880 (abc biopply, Solothurn, Switzerland) and the Ultra-Low Attachment microcavity plate (Corning Elplasia Plates, Corning Inc., Corning, NY, USA). Spheroids were evaluated using molecular biology analyses of the principal chondrogenic markers by investigating physical parameters such as the mass density, weight, and size of spheroids and by confocal imaging. We hypothesized that there was a correlation between spheroid chondrogenic differentiation and the measurement of physical parameters that could define a critical point at which they start to mature.

## 2. Materials and Methods

The experimental outline of the research work is shown in Figure 1.

### 2.1. Adipose Tissue-Derived Stromal Cells Isolation

ADSCs left over from a previous study approved by the Ethics Committee of Istituto Ortopedico Rizzoli (“ADIPO_CELL”) were anonymized and used in the present research. Cells were obtained from waste adipose tissue material collected from four patients (two males and two women, aged 60 ± 10 years) during knee regenerative medicine treatments. Briefly, adipose tissues were washed with phosphate-buffered saline (PBS), and cells were isolated using enzymatic digestion with 0.05% type I collagenase (Sigma-Aldrich, St. Louis, MO, USA) at 37 °C for 30 min. Then, cells of the Stromal Vascular Fraction were washed and seeded into culture flasks (20 × 10^3^ cells/cm^2^) with α-MEM (Sigma-Aldrich) containing 15% FBS. At confluence, ADSCs were detached with trypsin–EDTA and frozen in liquid nitrogen.

### 2.2. Chip/ULA Seeding

ADSCs were thawed and expanded for two passages before being seeded into 3D CoSeedis™ Chip880 (abc biopply, Solothurn, Switzerland) or in a 24-well Black/Clear Round Bottom Ultra-Low Attachment (ULA) microcavity plate (Elplasia Plates, Corning Inc., Corning, NY, USA). Before seeding ADSCs, chips were equilibrated according to the manufacturer’s protocol. Briefly, chips were put into a six wells plate containing 5 mL of growth medium (Dulbecco’s modified Eagle’s medium high glucose (DMEM, Sigma-Aldrich)) with 10% FBS, centrifuged for 3–5 min at 200 *g* to remove air bubbles, and equilibrated for 3 h in a 37 °C humidified incubator (chips take up the color of the medium). The equilibration medium was successively replaced with 9.5 mL of fresh medium containing 1 × 10^6^ ADSCs, and the seeded chips were transferred to the incubator for 24 h to allow cells to sediment completely and form 3D organoids. ULA wells were washed several times with medium before the seeding to remove air bubbles from the microcavity and allow ADSCs (6.3 × 10^5^/1.5 mL/wells) to sediment and assume a 3D architecture. One spheroid was obtained starting from about 1000 cells in both methods evaluated.

### 2.3. Chondrogenic Differentiation

For chondrogenic differentiation, ADSCs spheroids grown in both chip and ULA were cultured in growth medium for 24 h, which was then replaced with chondrogenic medium consisting of DMEM (Sigma-Aldrich) with 10% FBS, 100X ITS-Premix (BD Biosciences, Bedford, MA, USA), 10^–7^ dexamethasone (Sigma-Aldrich), 37.5 g/mL ascorbate-2 phosphate (Sigma–Aldrich), 1 mM of sodium pyruvate (Sigma-Aldrich), pen-streptomycin (100 U/mL 100 g/mL, Gibco, Grand Island, NY, USA), and 10 ng/mL of TGF-β1 (Miltenyi Biotec B.V. and Co. KG, Bergisch Gladbach, Germany). The medium was changed twice a week, and spheroids were evaluated at 7, 14, 21, and 28 days using gene expression analyses and immunohistochemistry.

### 2.4. Analysis of mRNAs Expression by Real-Time PCR

Spheroids obtained from both chip and ULA were analyzed using real-time PCR to investigate the expression of specific chondrogenic markers such as collagen type II, Sox-9, and aggrecan. To harvest spheroids, chips were flipped upside-down, put into a well containing PBS, and centrifuged at 300 *g* for 30 s. Successively, empty chips were discarded, and organoids were collected with a serological pipette. Spheroids were aspirated using a serological pipette from a ULA plate. To ensure no organoids were left behind in the wells, several washes with PBS were performed.

RNA was isolated using TRIzol reagent (Invitrogen) following the manufacturer’s instruction. After treatment with DNase I (DNA-free Kit; Ambion, Life Technologies, Carlsbad, CA, USA) and RNA quantification using a Nanodrop ^®^ spectrophotometer (EuroClone S.p.a.), 0.5 μg of RNA was reverse transcribed using MuLV reverse transcriptase (Thermo Fisher Scientific, Waltham, MA, USA). PCR primers for the selected genes and for the housekeeping gene glyceraldehyde-3-phosphate dehydrogenase (GAPDH) used as internal control are listed in Table 1. Real-time PCR was run with the following protocol: initial activation at 95 °C for 10 min, amplification for 45 cycles at 95 °C for 5 s and 60 °C for 20 s, in a LightCycler Instrument (Roche Molecular Biochemicals, Indianapolis, IN, USA) using SYBR Premix Ex Taq (Takara, Clontech Laboratories, Mountain View, CA, USA). mRNA levels were calculated for each target gene and normalized using the reference gene GAPDH according to the formula 2^−ΔCt^.

### 2.5. Clearing, Labeling, and Imaging of Spheroids

Spheroids were treated with PFA 4% for 3 h at room temperature (RT) and then rinsed with phosphate-buffered saline (PBS) before being stored at 4 °C. Next, spheroids were incubated with the X-CLARITY Hydrogel-Initiator solution from Logos Biosystems, Inc., Anyang-si, South Korea for 22 h at 4 °C following the manufacturer’s instructions. Afterward, spheroids were polymerized for three hours using the X-ClarityTM Polymerization System (Logos Biosystems, Anyang-si, South Korea), with the vacuum set at 90 kPa and a temperature of 37 °C. Spheroids were then washed with PBS and stored at 4 °C. The samples-hydrogel hybrid was then subjected to 8 h of treatment in the X-ClarityTM Tissue Clearing System from Logos Biosystems, with the current set at 0.8 A, a temperature of 37 °C, and a pump speed of 30 rpm. Finally, spheroids were washed several times with PBS to remove the cleaning solution and stored at 4 °C until they were ready to be used. To prevent nonspecific bindings, spheroids were first incubated with 3% bovine serum albumin (BSA) (Sigma-Aldrich, St. Louis, MO, USA) for one hour at room temperature. Then, spheroids were incubated overnight at 4 °C with a mouse polyclonal anti-Collagen type II antibody (Sigma-Aldrich), diluted to 1:30 in PBS 3% BSA. After one hour of washing with PBS 3% BSA, the samples were incubated with goat anti-mouse Cy5 conjugated antibody (Invitrogen) in PBS containing 3% BSA for 1 h. Spheroids were then washed with PBS and labeled with DAPI (Sigma-Aldrich, St. Louis, MO, USA) for 20 min at room temperature. After another washing step with PBS, the samples were mounted with a mixture of X-CLARITY mounting solution from Logos Biosystems and 1,4-Diazabicyclo [2.2.2] octane (DABCO) from Sigma-Aldrich. The fluorescent images of the clarified spheroids were visualized and imaged using a Nikon A1-R confocal laser scanning microscope equipped with a 25x (silicon immersion, 1.05 NA) apochromatic objective and with 405 and 647 laser lines to excite DAPI and Cy5 fluorescence signals. Z-stacks were collected at an optical resolution of 500 nm/pixel, stored at 12-bit with 4096 different gray levels, pinhole diameter set to 1 Airy unit, and z-step size set to 1 μm. The data acquisition parameters, such as laser power, gain in amplifier, and offset level, were set up in a fixed manner. Confocal images were processed using the Richardson-Lucy deconvolution algorithm. Finally, volume measurements, fluorescence quantification, and the volume view with 3D rendering were carried out using the NIS Elements Advanced Research software AR 5.20 from Nikon Instruments, Tokyo, Japan. A minimum of 10 single spheroids were analyzed for every tested condition and performed in triplicate.

### 2.6. Biophysical Characterization of Mass Density, Weight, and Diameter using the W8 Physical Cytometer

To assess the biophysical characterization of spheroids, a novel instrument that enables precise, simultaneous, and rapid quantification of mass density, weight, and size of spheroids, the W8 Physical Cytometer (CellDynamics iSRL, Bologna, Italy), was used. At the set time points (7, 14, 21, 28 days), spheroids were fixed with PFA 4% at room temperature for 3 h and washed twice with Dulbecco’s phosphate-buffered saline (DPBS) 1X w/o Ca^2+^ and Mg^2+^ (Corning Life Sciences, Durham, NC, USA). Before the measurements with the W8 Physical Cytometer (CellDynamics), all the samples were resuspended in 7 mL of W8 Analysis Solution (W8AS, CellDynamics) and analyzed according to the previously described procedure [28]. At least 10 spheroids for each condition were analyzed in triplicate. For each sample, values were obtained from two measurement repetitions. 

### 2.7. Min-Max Normalization of Biophysical Parameters and Gene Expression

Min-max normalization was applied to compare biophysical parameters and gene expression. Briefly, this method is a data preprocessing technique commonly used to scale and standardize numerical values within a specific range. The purpose is to transform the original data so that it falls within a consistent interval, typically between 0 and 1. This is achieved by subtracting the minimum value of the dataset from each individual data point and then dividing by the range of the data (the difference between the maximum and minimum values).
Min-max normalization:
normalized_value = (original_value − min_value)/(max_value − min_value)

The obtained normalized data were then expressed in percentages to ensure that all the values in the dataset were proportionally scaled to fit within the specified range, making it easier to compare and analyze different datasets that may have different scales. The original min and max values of the formula are based on the mean value obtained from the measurement repetition of each individual data point.

### 2.8. Statistical Analysis

Shapiro–Wilk test was first performed on the obtained mass density, weight, and diameter outputs to analyze the distribution of the dataset based on skewness and/or kurtosis, as previously reported [28]. For all the cases of non-normal distribution, the Tukey method was then carried out to identify and eliminate outliers (K > 1.5). Subsequently, the Shapiro–Wilk approach was reiterated to confirm the normal distribution. Data are presented as mean ± SD. Statistical analysis was performed using one-tailed and two-tailed unpaired Student’s *t*-test. The cut-off value of significance is indicated in each figure legend.

## 3. Results

### 3.1. Gene Expression

The expression of Sox-9, Collagen II, and aggrecan was examined using real-time PCR at 7, 14, 21, and 28 days. Data indicated that the expression of Sox-9, a transcription factor expressed at the early stage of chondrogenesis, showed an upregulation from day 7 up to day 21 and decreased at day 28 in both chips and ULA groups (Figure 2A). Collagen type II significantly increased from day 7 up to day 21 and day 28 (*p* < 0.05) in cells cultured into chips. At the same time, no differences were observed in ADSCs grown in ULA plates. Furthermore, a higher gene expression for collagen type II was detected on day 28 in chip-cultured cells compared to the ULA ones (*p* < 0.05) (Figure 2B).

Aggrecan, one of the most important proteoglycans of cartilage tissue, displayed the same trend in both chips and ULA-derived spheroids. It increased from day 7 to day 28; however, no differences between the two groups analyzed were observed (Figure 2C).

### 3.2. Collagen Type II Immunohistochemistry

By confocal microscopy, we analyzed the 3D cell architecture of spheroids cultured for 21 and 28 days on-chip or in ULA plates. We imaged cleared spheroids combined with a silicone immersion oil objective suitable for deep imaging analysis as previously described [27]. The cultured spheroids on-chip appeared smaller in size (as shown in Figure 3A,B) after 21 and 28 days, which could be due to the small size of the culture wells. However, no significant structural differences were observed between the spheroids cultured in ULA plates and on-chip (as seen in Figure 3A). On the other hand, examining the expression of Collagen type II, we observed that the fluorescence intensity increased in the cultured spheroids on-chip after 28 days (as shown in Figure 3A,C), indicating improved spheroid differentiation. Furthermore, the comparison of spheroids grown on chips and those grown in Ultra-Low Attachment plates highlighted that the culture method significantly influences the maturation trajectory. Spheroids generated with chips displayed a more advanced state of maturation, as evidenced by higher gene expression of collagen type II and increased fluorescence intensity for this marker in confocal microscopy.

### 3.3. Biophysical Characterization of Spheroids

The biophysical characterization of chondrogenic differentiation in the two different systems showed that weight and diameter have comparable trends over time. In fact, both parameters declined progressively and significantly over time, from 7 to 28 days, in both chip and ULA-grown spheroids (Figure 4B,C). However, the trend of mass density displayed differences between the two growth systems (Figure 4A). Indeed, mass density progressively increased, reaching a plateau at 21 days on spheroids growth on a chip, while on samples seeded on ULA, mass density significantly decreased after 14 days and then enhanced in the following 7 days, arriving again at a plateau at 21 days.

The analysis of the two groups indicated notable variations in mass density across all time points examined. Notably, spheroids subjected to chondrogenic differentiation on chips consistently demonstrated greater mass density compared to samples cultivated in ULA, with statistical significance observed at 14 and 21 days (*p* < 0.001) and a trend towards significance at 28 days (*p* < 0.1), except for the 7-day time point. 

### 3.4. Data Comparison of Biophysical Parameters and Gene Expression

To compare biophysical characterization data with gene expression profiles, the statistical tool of min-max normalization was used. This method allowed the comparison over time of gene expression and biophysical parameters, highlighting the mutual evolutions and fluctuations of the parameters under analysis. This normalization technique ensured that both gene expression and biophysical data were brought to a common scale, allowing for a comprehensive analysis of their dynamic evolution over time (Figure 5).

As illustrated in Figure 5A–C, spheroids grown on chips show a parallel reduction in weight and diameter over time. When comparing this pattern with gene expression, a distinct trend emerges among the genes examined. Gene expression is reduced at 7 days, followed by a gradual increase on subsequent days, with specific gene-dependent increases. Noteworthy in spheroids grown in chips is a concomitance between the progression of gene expression and changes in mass density, indicating that mass density is the most indicative biophysical parameter during chondrogenic differentiation. Conversely, when examining the ULA-grown spheroids (as illustrated in Figure 5D,E), the changes in weight and diameter show a similar trend to that observed in the chips. However, in terms of mass density, its increase over time is less indicative of the progression of gene expression and of an advanced state of chondrogenic maturation. Additionally, the correlation between mass density and gene expression suggests that mass density serves as a valuable indicator of spheroid maturation status, providing insights into the chondrogenic differentiation process.

## 4. Discussion

Three-dimensional cell cultures better mimic the physiological environment and tissue-specific cellular behavior than traditional monolayer cultures and are becoming a useful tool in several research fields, such as cancer cell biology, drug screening, tissue engineering, and stem cell research [20,29]. In recent years, the use of spheroids derived from different cell types has gained significant attention in the field of regenerative medicine and cartilage repair [30,31]. Based on the specific research purposes, spheroids can be obtained with several methods that are able to provide specific geometry and architecture, essential to create models that more closely mimic native tissue environments. In this study, we investigated the self-assembled capacity of ADSCs to form 3D cartilage microtissues using two non-adhesive microwell systems (3D chips vs. Ultra-Low Attachment plates). These high-throughput techniques allowed the controlled formation of spheroids, preventing their loss due to the floatation phenomenon, resulting in a specific geometry and size after 28 days of culture. We analyzed spheroids obtained with the two systems, comparing the chondrogenic potential of ADSCs at both gene and protein levels and by the measurement of spheroids’ mass density, weight, and diameters.

The data of this study highlighted that ADSC-derived spheroids, grown in both chip and ULA systems, were able to express the typical genes involved in chondrogenic differentiation. These findings are in line with Tsvetkova et al., who demonstrate that MSC from adipose tissue grown as spheroids exhibited maximum chondrogenic potential compared to MSC derived from other sources [32]. Despite the similar gene expression of Sox-9 and Aggrecan, a higher expression of Collagen type II (the main marker of hyaline cartilage) was evident in spheroids obtained from the chip, indicating that this culture method improved the chondrogenic capacity of the cells. This is probably due to the conical shape of the chip microarchitecture, which can provide a well-defined geometry with high cell–cell interaction and paracrine signaling that could guide the formation of cartilage-like tissues. This finding was confirmed using immunohistochemistry, which highlighted a smaller size and a greater production of Collagen II in spheroids grown in the chip system at both 21 and 28 days compared to the ULA one.

To obtain a measurement of fluorescence intensity and spheroid volume, we analyzed the whole spheroids using microscopy techniques capable of penetrating the depth of complex and highly light-scattering samples, avoiding conventional techniques that require sample embedding and cutting. Histology/immunohistochemistry applied to cellular spheroids has several drawbacks, such as deformation and fracture of the spheroids, poor contrast of conventional stains, and low spatial resolution. Moreover, confocal fluorescence microscopy has limited penetration depth and cannot penetrate more than a few cellular layers into the spheroid [33]. To overcome these issues, a method that combines a clearing technique and a silicone oil objective able to remove the lipid component from the sample and reduce spherical aberration was optimized [27]. This approach allowed confocal microscopy to achieve good light penetration without introducing high levels of spherical aberration, providing an accurate measure of fluorescent intensity and spheroid volume.

The biophysical characterization of spheroids during chondrogenic differentiation provided interesting information. According to confocal analyses, the size and the weight of the spheroids decreased over time in both systems and if compared at the final differentiation time, no significant differences were observed between chips and ULAs. Therefore, based solely on size and weight evaluation, the conclusion would be that chondrogenic differentiation is similar in the two supports used. However, when cross-referencing this result with the gene expression of the examined markers and collagen expression in confocal microscopy, it is evident that differentiation is more pronounced in chips compared to ULAs. This divergent trend in the two supports may be correlated with variations in mass density. In fact, in the chips at 28 days, the mass density is higher compared to ULAs, and as reported in the literature for perinatal stem cells [25], an increase in mass density is associated with the secretion of different ECM proteins within the spheroid formation. The correlation between mass density and gene expression was confirmed using the statistical analysis, suggesting that mass density stands out as the most indicative biophysical parameter in the context of chondrogenic differentiation, and it could, therefore, represent a crucial point during spheroid maturation.

This research presents some limitations. Primarily, the restricted sample size hindered the possibility of conducting further analyses. Additionally, it is important to note that the ADSCs utilized were derived from human primary cultures, thereby introducing an added layer of experimental variability. Consequently, it would be advisable to expand the scope of evaluations to encompass a larger sample size. This expansion would help affirm the presented findings and offer greater insight into potential factors that affect the chondrogenic differentiation process and its interplay with mass density. However, despite these shortcomings, this study managed to derive significant conclusions related to its primary objective, unveiling the importance of evaluating mass density in the study of 3D model evolution.

The data presented suggest that the maturation process is influenced by the culture method, with spheroids grown on chips showing enhanced chondrogenic differentiation compared to those grown in Ultra-Low Attachment plates. In the context of three-dimensional cell cultures, particularly with the growing interest in spheroids, the precise manipulation of spheroid dimensions, contour, and architecture becomes crucial for replicating natural tissue settings. Small spheroids, which exhibit adaptability to custom-designed molds and hold promise for bioprinting applications, open up exciting opportunities for building larger-scale tissue constructs with predefined forms. Additionally, their compatibility with bioassembly techniques allows for the generation of intricate tissue structures using micro-precursor components, thereby driving progress in the fields of cartilage repair and regenerative medicine. The critical maturation point is characterized by increased gene expression of key cartilaginous markers, such as collagen type II, and is closely associated with changes in mass density. This suggests that mass density is a significant biophysical parameter indicative of spheroid maturation. Therefore, the discussion of maturation revolves around the idea that spheroids can progress along a chondrogenic pathway, and this progression can be tracked and influenced by cultural conditions.

## 5. Conclusions

This study highlighted that ADSCs can form spheroids that exhibit the capability to differentiate in a chondrogenic sense using the expression of cartilaginous markers. Moreover, we demonstrated a direct correlation between mass density and spheroids’ differentiation, establishing a crucial stage signifying their initiation of maturation.

This study sheds light on the potential of adipose stromal cell spheroids as a promising tool for cartilage repair. It emphasizes the importance of considering the concept of maturation in the context of spheroid-based tissue engineering. The results suggest that the culture method, such as using 3D chips, can enhance the chondrogenic differentiation of spheroids, leading to improved cartilage-like tissue formation. Furthermore, the study underscores the critical role of mass density as a key biophysical parameter that correlates with spheroid maturation. These findings have significant implications for the development of regenerative therapies. By better understanding and controlling the maturation process of spheroids, we can potentially advance the field of cartilage regeneration and tissue engineering for personalized therapeutic intervention.

## Figures and Tables

**Figure 1 bioengineering-10-01182-f001:**
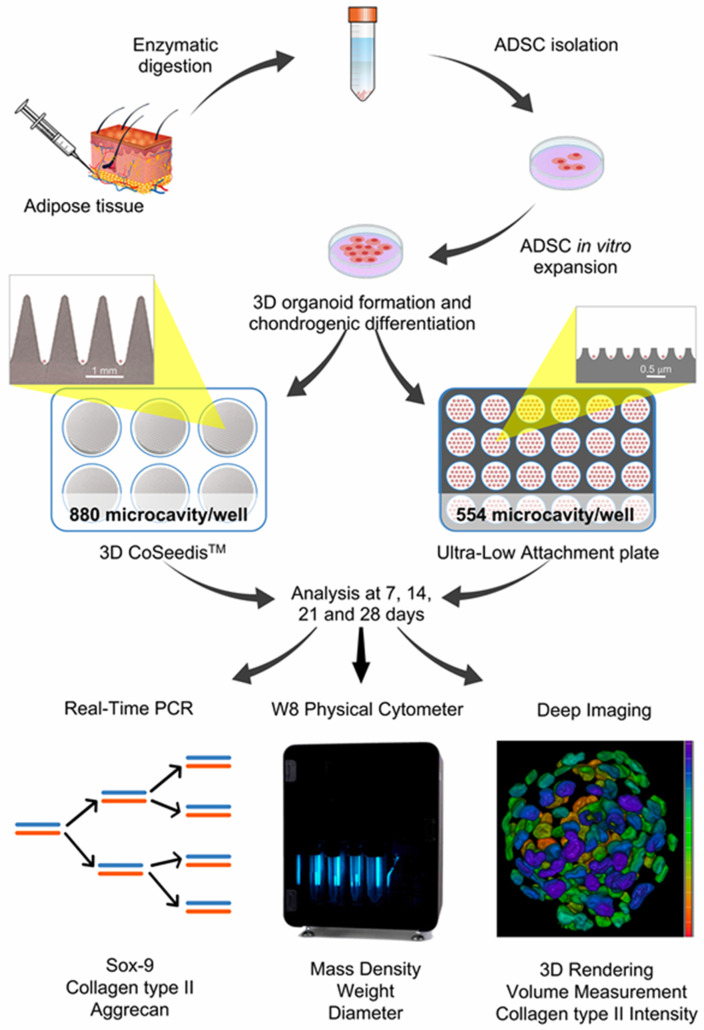
Overall experimental procedure from ADSCs isolation and spheroids formation to molecular biology/biophysical parameters/deep imaging evaluations.

**Figure 2 bioengineering-10-01182-f002:**
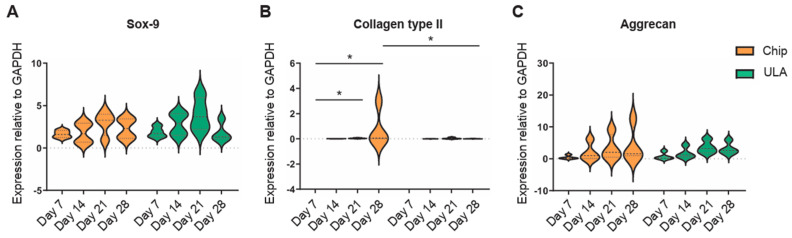
Violin plots showed gene expression at 7, 14, 21, and 28 days of the principal cartilaginous markers (**A**) Sox-9, (**B**) Collagen type II, and (**C**) Aggrecan. Data were normalized to GAPDH and analyzed using Student’s *t*-test for two-group comparisons with * *p* < 0.05. Different patterns were used for different spheroids culture methods: chips orange and ULA green.

**Figure 3 bioengineering-10-01182-f003:**
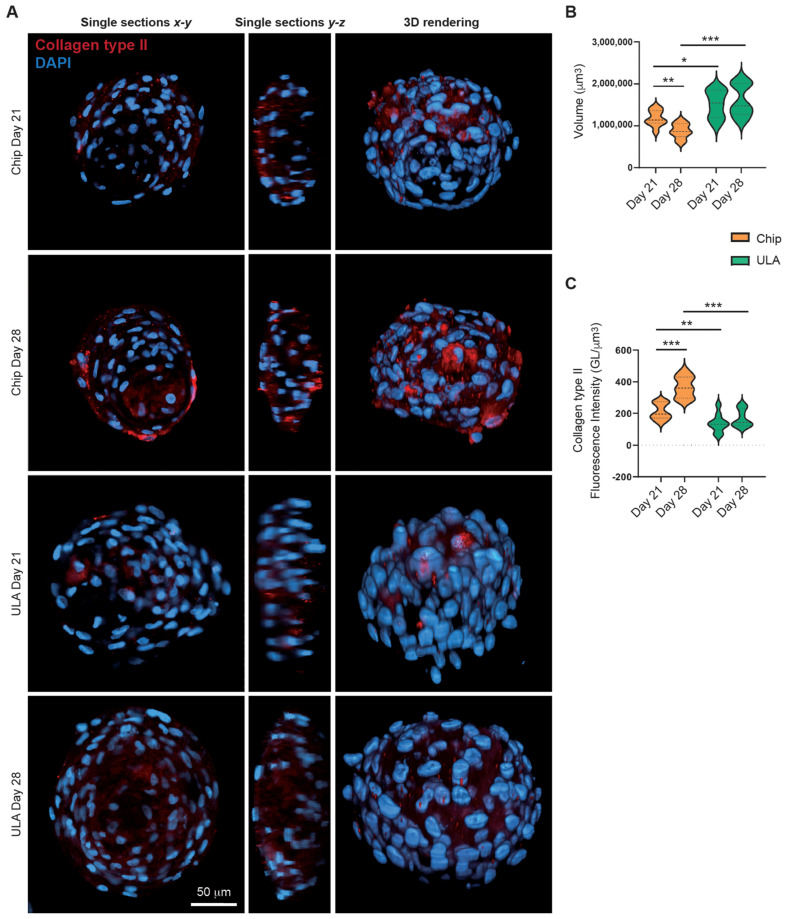
(**A**) Representative images of spheroids cultured for 21 and 28 days on-chip or in ULA plates. The clarified spheroids were analyzed using confocal microscopy. The spheroids were stained with DAPI (blue) and anti-Collagen type II antibody (red). In the left panels, x-y single optical sections from the maximum diameter of the spheroids are shown, scale bars 50 µm. In the central panel, z-y cross sections are represented. The 3D rendering projections are focused on the right panels. (**B**) Spheroids volume after 21 or 28 days of differentiation on-chip or in ULA plates. Statistical analysis was performed using a two-tailed unpaired Student’s *t*-test. * *p* < 0.05, ** *p* < 0.01, and *** *p* < 0.001. (**C**) Mean Fluorescence Intensity of anti-collagen type II antibody in spheroids cultured for 21 and 28 days on-chip or in ULA plates. The measurement is calculated in each optical section from the sum of the values of all the pixels divided by the number of voxels occupied by the spheroid. Statistical analysis was performed using two-tailed unpaired Student’s *t*-test. * *p* < 0.05, ** *p* < 0.01, and *** *p* < 0.001.

**Figure 4 bioengineering-10-01182-f004:**
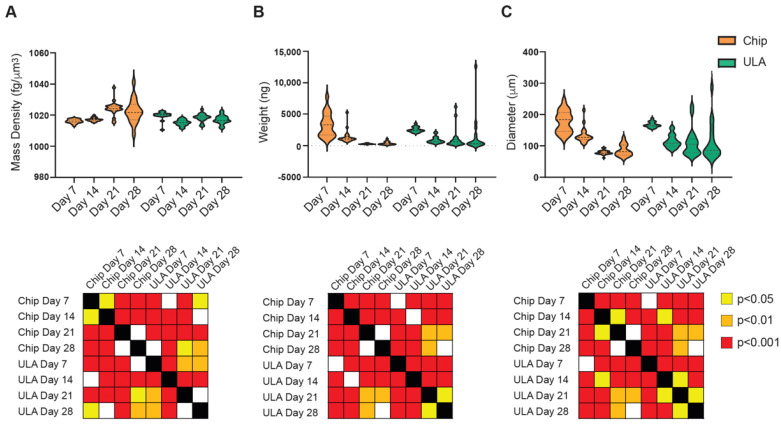
On the top: violin plots showed measurement of mass density (**A**), weight (**B**), and diameter (**C**) of ADSCs spheroids cultured in ULA plates (green violins) or on-chip (orange violins) at 7, 14, 21, and 28 days of culture; on the bottom: statistical analysis performed using two-tailed unpaired Student’s *t*-test. The grid representation using colored squares corresponds as follows: white indicates no significance, yellow indicates *p* < 0.05, orange indicates *p* < 0.01, and red indicates *p* < 0.001.

**Figure 5 bioengineering-10-01182-f005:**
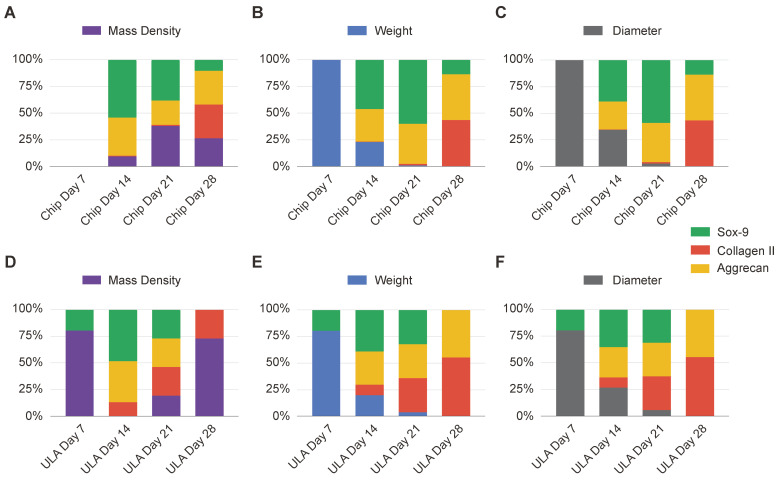
Min-Max normalization of mass density ((**A,D**), depicted in violet), weight ((**B,E**), represented in blue), and diameter ((**C,F**), shown in grey) in relation to the gene expression levels of Sox-9 (shown in green), collagen type II (depicted in red), and aggrecan (displayed in yellow). The Min-Max normalization was computed over time for chips (**A**–**C**) and ULA (**D**–**F**). The values were expressed as a percentage of the data range.

**Table 1 bioengineering-10-01182-t001:** List of primers used in real-time PCR.

RNA Template	Primer Sequences (5′-3)	Annealing Temperature (°C)
GAPDH	5′-TGG TAT CGT GGA AGG ACT CAT GAC 3′-ATG CCA GTG AGC TTC CCG TTC AGC	60
Collagen type II	5′-GAC AAT CTG GCT CCC AAC 3′-ACA GTC TTG CCC CAC TTA C	60
Aggrecan	5′-TCG AGG ACA GCG AGGCC 3′-TCG AGG GTG TAG CGT GTA GAGA	60
Sox-9	5′-GAG CAG ACG CAC ATCTC 3′-CCT GGG ATT GCC CCGA	60

## Data Availability

Not applicable.

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
