# Peer review of "Adipose Stromal Cell Spheroids for Cartilage Repair: A Promising Tool for Unveiling the Critical Maturation Point"

_bioengineering, 2023, doi:10.3390/bioengineering10101182_

Round 1

Reviewer 1 Report

This study was to evaluate the capacity of human ADSCs to assemble in spheroids and to differentiate into chondrogenic lineage from the perspective of their use for cartilage repair. Spheroids were obtained by means of two different cultured methods and evaluated by molecular biology analyses of the principal chondrogenic markers, by the investigation of physical parameters such as the mass density, weight, and size of spheroids, and by confocal imaging. The results suggest that the culture method, such as using 3D chips, can enhance the chondrogenic differentiation of spheroids, leading to improved cartilage-like tissue formation. Furthermore, the study underscores the critical role of mass density as a key biophysical parameter that correlates with spheroid maturation. These findings have significant implications for the development of regenerative therapies and tissue engineering for personalized therapeutic intervention using cellular spheroids. There are some comments:

P4 L123 – It is not indicated how many cells were in one spheroid

P8 L266 - "21" is missing.

Author Response

We thank the reviewer for your positive comments.

P4 L123- We add the information at P4 L127: “One spheroid was obtained starting from about 1000 cells in both methods evaluated. “

P8 L266- we inserted the missing day.

Reviewer 2 Report

This is a preclinical trial for developing cartilage from human adipose stroll cells by 3D and ultra low attachment cultures. While the size and volume were higher in cellular spheroids constructed by ultra low attachment culture. Expression of collagen was richer by 3D culture. That means the characteristics of cartilage spheroids were different by the culture methods. It will contribute to formation of ideal cartilage which can be available to clinical setting.

Please discuss following three simple questions.

1. Why the difference was occurred by the different culture methods?

2. The formed spheroids were too small. How will the author form cartilage from the spheroids?

3. The therapeutic effects of these spheroids are important for future clinical setting. How about plan to perform animal transplantation in this series?

Author Response

We thank the reviewer for the interesting questions.

  1. Why the difference was occurred by the different culture methods?

Culture methods can significantly influence spheroids formation and their differentiation capacity by creating a microenvironment that impacts cell behavior and interactions. Here we used two different approaches to obtain spheroids trying to standardize the culture methods in terms of cell type and source, cell density, medium composition, and culture duration. As claimed in the discussion, we hypothesize that the differences observed in spheroids maturation could be due to the different geometry of the chips compared to ULA, the conical shape of chip microarchitecture, could offer a more defined geometry facilitating cell-cell interaction and paracrine signaling. Moreover, the choice of culture substrates such as chip or ULA and their different surface properties can influence cell attachment and maturation, leading to varying effects on spheroid size, morphology, and differentiation.

  1. The formed spheroids were too small. How will the author form cartilage from the spheroids?

We agree with the reviewer that the formed spheroids are very small. Despite this, in the perspective of using spheroids for cartilage repair, over the past few decades, various methods have been explored for creating engineered cartilage tissue constructs. One innovative technology on the horizon is assembly, which refers to combining small biounits such as microtissues (cartilage micro-fragments) or micro-precursor tissues (spheroids or organoids) as fundamental building blocks for constructing larger-scale tissues. In this overview, we could speculate that small spheroids are the ideal bloks for the practical applications of bioassembly technology in the context of articular cartilage regeneration.

Thanks to their small size, spheroids can easily fit into an anatomically designed mold, allowing for the bioassembly of macro-scale tissues with predetermined shapes, including cylinders, circles, squares, and triangles.

Moreover, the use of small spheroids could be employed for bioprinting. Bioprinting processes utilizing bioinks loaded with spheroids provide a precise deposition of uniformly spherical cell aggregates in predetermined positions through a controlled and reproducible method, obtaining a structure with a well-defined geometry.

We add this concept at P12 L403:

“In the context of three-dimensional cell cultures, particularly with the growing interest in spheroids, the precise manipulation of spheroid dimensions, contour, and architecture becomes crucial for replicating natural tissue settings. Small spheroids, which exhibit adaptability to custom-designed molds and hold promise for bioprinting applications, open up exciting opportunities for building larger-scale tissue constructs with predefined forms. Additionally, their compatibility with bioassembly techniques allows for the assembly of intricate tissue structures using micro-precursor components, thereby driving progress in the fields of cartilage repair and regenerative medicine.”

  1. The therapeutic effects of these spheroids are important for future clinical setting. How about plan to perform animal transplantation in this series?

Animal transplantation studies are a critical step in translational research, providing valuable insights into the safety and efficacy of potential therapies before moving to human clinical trials.

However, at this stage, animal transplantation is far from being achieved. Even if we demonstrated that 3D chips are a good choice to form spheroids able to differentiate into chondrogenic potential, there are other aspects that should be taken into account. It's important to consider that the specific procedures and protocols may vary depending on the type of animal model, the research goals, and the intended clinical applications. Additionally, ethical considerations and regulatory approvals should be addressed when conducting experiments involving animals.

Reviewer 3 Report

This is a good and systematic study. I am only concerned about Figure 5. Will it be possible to provide a description of SD? 

Regarding cartilage regeneration, please cite this reference for comparison on current progress in therapy without cells:

Amsar RM, Wijaya CH, Ana ID, Hidajah AC, Notobroto HB, Kencana Wungu TD, Barlian A. Extracellular vesicles: a promising cell-free therapy for cartilage repair. Future Sci OA. 2021 Dec 6;8(2):FSO774. doi: 10.2144/fsoa-2021-0096. PMID: 35070356; PMCID: PMC8765097.

The language is ok. But if the authors can provide language editing certificate, it will be better. 

Author Response

We thank the Reviewer for this insightful comment which helped us to better explain the normalization applied.

Min-max normalization is a method applied to the original dataset and do not take into account the standard deviation of the values as the relevance of the obtained dataset was already proven by the statistical analyses performed and expressed in fig 2 for gene expression and fig 4 for biophysical characterization.

We better underlined the Min-max normalization method in the manuscript at P6 L217

“Min-max normalization:

normalized_value = (original_value - min_value) / (max_value - min_value)

The obtained normalized data were then expressed in percentage to ensures that all the values in the dataset are proportionally scaled to fit within the specified range, making it easier to compare and analyze different datasets that may have different scales.

Original-min and max values of the formula are based on the mean value obtained from the measurement repetition of each individual data points.”

We also correct the indications of Figure 5 at P10 L319 and P10 L326

Regarding cartilage regeneration, please cite this reference for comparison on current progress in therapy without cells:

We add this reference at P1 L46 [8]

The language is ok. But if the authors can provide language editing certificate, it will be better. 

We were unable to utilize the journal's language editing service because our Institute couldn't make immediate payments; consequently, we enlisted the assistance of a native speaker to review the paper at L33, L56, L6, L66, L78, L79, L85, L91-93, L244, L289, L290, L323, L330, L332, L334, L342, L347, L362, L363, L366, L374, L375.